# THE WORLD IS NOT MONO: ENABLING SPATIAL UNDERSTANDING IN LARGE AUDIO-LANGUAGE MODELS

## ABSTRACT

Existing large audio-language models perceive the world as "mono"—a single stream of audio that ignores the critical spatial dimension ("where") required for universal acoustic scene analysis. To bridge this gap, we first introduce a hierarchical framework for Auditory Scene Analysis (ASA). Guided by this framework, we introduce a system that enables models like Qwen2-Audio to understand and reason about the complex acoustic world. Our framework achieves this through three core contributions: First, we build a large-scale, synthesized binaural audio dataset to provide the rich spatial cues. Second, we design a hybrid feature projector, which leverages parallel semantic and spatial encoders to extract decoupled representations. These distinct streams are integrated via a dense fusion mechanism, ensuring the model receives a holistic view of the acoustic scene. Finally, we employ a progressive training curriculum, advancing from supervised fine-tuning (SFT) to reinforcement learning via Group Relative Policy Optimization (GRPO), to explicitly evolve the model's capabilities towards reasoning. On our comprehensive benchmark, the model demonstrates comparatively strong capability for spatial understanding. By enabling this spatial perception, our work provides a clear pathway for leveraging the powerful reasoning abilities of large models towards holistic acoustic scene analysis, advancing from "mono" semantic recognition to spatial intelligence.

## 1 INTRODUCTION

The expansion of Large Language Models (LLMs) from text-centric processing to multimodal intelligence has brought renewed attention to the auditory domain. Recent Large Audio-Language Models (LALMs) have demonstrated remarkable capabilities in understanding and following instructions related to the semantic content of audio—the 'what' in speech, general sounds, and music (Chu et al., 2024; Yang et al., 2024). However, prevailing LALM frameworks predominantly treat audio as a monophonic time series, lacking explicit modeling of spatial attributes—the 'where'. Technical reports and public benchmarks for these models rarely address reasoning about direction, distance, or spatial relationships, a gap that constitutes the primary motivation for our work.

To formalize and systematically address this gap, we introduce a novel three-layer cognitive framework for Auditory Scene Analysis (ASA). Our framework is inspired by classic cognitive models in the field of Computational Auditory Scene Analysis (CASA), particularly Albert Bregman's two-stage theory of human auditory perception, which distinguishes between data-driven "Primitive Grouping" and knowledge-driven "Schema-based Grouping" (Bregman, 1990). We propose that genuine auditory intelligence requires progressing through three sequential levels of capability:

- **L1: Static Identification**: The foundational perceptual layer, responsible for identifying the discrete properties of individual sounds (e.g., their semantic class, 'what', and their spatial location, 'where').
- **L2: Relational Integration**: The relational aggregation layer, responsible for associating or "binding" these disparate attributes into a single, coherent auditory object (e.g., knowing *that specific* 'dog bark' came from *that specific* 'left-hand' location).
- **L3: Cognitive Reasoning**: The high-level cognitive layer, which uses the bound objects from L2 and world knowledge ('schemas') to infer complex relationships, causality, and scene context.

Using this framework, we can clarify the current limitation: existing LALMs are largely confined to the *semantic-half* of L1, while almost completely lacking the *spatial-half* of L1, as well as L2 and L3 capabilities.

This disregard for spatial dimensions starkly contrasts with the fundamental mechanisms of biological hearing. As established in classical auditory scene analysis, the brain leverages binaural cues, such as Interaural Time Differences (ITD) and Interaural Level Differences (ILD), to localize sound sources and segregate auditory streams. This process provides a "spatial release from masking" and underpins the 'cocktail party effect,' enabling focused listening in complex acoustic environments. A model incapable of processing these binaural cues is, in a perceptual sense, "spatially deaf," responding only to a one-dimensional projection of a three-dimensional sound field.

This limitation imposes significant constraints on real-world applications in robotics, augmented reality, and assistive hearing, all demanding a joint understanding of audio's content and spatial arrangement. Recent explorations have begun extending the "encoder-to-LLM" paradigm to spatial audio, typically by processing binaural signals or Ambisonics recordings (You et al., 2025). For instance, BAT(Zheng et al., 2025) introduced spatial question answering, while subsequent methods improved reasoning by decoupling spatial features or using contrastive learning for localization (Devnani et al., 2024). Despite these promising first steps, the research landscape remains fractured, with most large-scale models focused on semantics (Tang et al., 2024; Huang et al., 2023) while specialized systems handle spatial physics (Shimada et al., 2021). We propose to bridge this divide, using the reasoning core of an LLM to unify these capabilities and process open-ended spatial queries with physically consistent representations.

However, the primary bottleneck to integrating spatial awareness into general auditory intelligence is not the lack of algorithmic starting points, but the scarcity of appropriate data and representations. Large-scale audio datasets like FSD50K (Fonseca et al., 2022), Clotho (Drossos et al., 2019), and MusicCaps (Agostinelli et al., 2023b) are rich in semantic labels but lack the binaural or 3D spatial metadata required for joint semantic-spatial alignment. Real-world spatial data, while valuable, remains limited in scale and diversity. Consequently, physically-consistent procedural simulation emerges as the most viable path forward. Tools such as Pyroomacoustics (Scheibler et al., 2017) and high-resolution Head-Related Transfer Function (HRTF) databases (Audio Engineering Society, 2015) enable the systematic generation of large-scale training corpora with complete "position-semantic-environment" annotations.

In this context, we introduce **The World is Not Mono (TWNM)**, a framework that extends LALMs from semantic recognition to spatial reasoning by integrating synthetic binaural data, specialized representation learning, and reinforcement learning alignment.

**Contributions.** The main contributions of this work are:

- **Synthetic binaural data pipeline**: construction of a scalable simulation framework using physically realistic Binaural Room Impulse Responses (BRIRs) / Head-Related Transfer Functions (HRTFs) to generate large-scale, spatially annotated audio–language QA pairs across diverse environments.

- **Hybrid Feature Projector**: design of a dual-stream projection framework that explicitly decouples semantic and spatial processing using specialized expert layers, integrated via a dense fusion mechanism for feature alignment.

- **Training and alignment**: a progressive curriculum that first optimizes the projector and expert layers with the LLM frozen, followed by joint fine-tuning. To further align spatial reasoning with natural language, GRPO (Shao et al., 2024) is employed, using rule-based rewards and relative advantages to improve output accuracy in multiple-choice spatial QA tasks.

- **Comprehensive evaluation**: demonstration of robust spatial understanding and relational reasoning on our promoted benchmarks, highlighting a viable path for spatially aware LALMs.

This work positions spatial perception not as an incremental feature, but as a fundamental dimension essential for achieving genuine auditory intelligence. The remainder of the paper is organized as follows: Section 2 reviews prior efforts on LALMs, spatial audio modeling, and data synthesis. Section 3 shows the problem formulation of ASA. Section 4 introduces our proposed framework. Section 5 presents evaluation results on our proposed benchmark. Finally, Section 6 concludes with discussions and future directions.

## 2 RELATED WORK

**Large Audio-Language Models**  Foundation models such as Qwen2-Audio (Chu et al., 2024), SALMONN (Tang et al., 2024), and AudioGPT (Huang et al., 2023) unify speech, sound, and music under a language interface, but remain focused on semantic tasks. Most benchmarks neglect localization and spatial reasoning, leaving models unable to process direction or distance.

**Spatial Audio and LLMs**  Recent efforts extend the encoder-to-LLM paradigm to spatial audio. BAT(Zheng et al., 2025) evaluated binaural reasoning on the SpatialSoundQA benchmark, and ELSA aligned spatial audio and text via contrastive learning (Devnani et al., 2024). These approaches show potential but leave representation design and training strategies unresolved.

**Computational Auditory Scene Analysis (CASA)**  SELD methods using FOA input, e.g., ACC-DOA (Shimada et al., 2021) and STARSS23 (Shimada et al., 2023), jointly predict sound class and direction. While effective for structured outputs, they mismatch the open-vocabulary and conversational style of LALMs, and early QA adaptations remain limited.

**Data Synthesis**  Because large-scale spatially annotated corpora are scarce, benchmarks often rely on simulation. SpatialSoundQA, for example, renders binaural or FOA mixtures by convolving clean sources with room responses. Public content sets such as FSD50K (Fonseca et al., 2022), Clotho (Drossos et al., 2019), and MusicCaps (Agostinelli et al., 2023b), combined with tools such as Pyroomacoustics (Scheibler et al., 2017) and standardized HRTFs (Audio Engineering Society, 2015), provide reproducible pipelines, though still narrower in scope than large semantic datasets.

**Training Paradigms**  Scaling and alignment methods from general-purpose LLMs also inform spatial modeling. MoE architectures (Fedus et al., 2022; Lepikhin et al., 2020) allow efficient specialization, while reinforcement learning techniques such as GRPO (Shao et al., 2024) refine reasoning and output formats. However, their application to spatial audio remains at an early stage.

## 3 TASK DEFINITION OF AUDITORY SCENE ANALYSIS

We define ASA by deconstructing the task into three distinct layers: Static Identification ($\mathcal{L}_1$), Relational Quantification ($\mathcal{L}_2$), and Cognitive Reasoning ($\mathcal{L}_3$). This framework aligns with classical theoretical research in auditory scene analysis and aims to provide guidance for the construction of machine auditory frameworks.

### 3.1 PRELIMINARIES AND DEFINITIONS

We define an auditory scene $\mathcal{S}$ as a composition of atomic units governed by physical laws and semantic rules.

**Objects ($\mathcal{O}$):** Let $\mathcal{O} = \{o_1, \ldots, o_N\}$ be the set of perceived auditory objects. Each object $o_i$ is defined by a disentangled attribute tuple:

$$o_i = (c_i, \tau_i, \mathbf{s}_i)$$

where $c_i \in \mathcal{C}$ denotes the semantic class (distinguishing foreground events $o^e$ from background $o^{env}$), $\tau_i = [t_{start}, t_{end}] \in \mathbb{R}^2$ denotes the temporal span, and $\mathbf{s}_i = (\phi, \theta, d) \in \mathbb{R}^3$ represents spatial coordinates.

**Knowledge Base ($\mathcal{K}$):** We assume access to a global knowledge base $\mathcal{K}$ containing physical laws, schemas, and causal graphs, serving as priors for high-level reasoning.

### 3.2 LAYER 1: STATIC IDENTIFICATION ($\mathcal{L}_1$)

The foundational layer $\mathcal{L}_1$ targets atomic perception, mapping the raw waveform $\mathbf{x} \in \mathbb{R}^T$ to a set of discrete, statistically independent entities without inferring higher-order dependencies. Formally, the output is:

$$\mathcal{O}_{\mathcal{L}_1} = \{o_1, o_2, \ldots, o_N, o_{env}\}$$

This layer executes attribute extraction across three dimensions: **Semantic Classification** ($c$) maps segments to labels, distinguishing discrete sound actions ($o^e$) from global scenes ($o^{env}$); **Temporal Segmentation** ($\tau$) localizes energy boundaries to define existence intervals $[t_{\text{start}}, t_{\text{end}}]$; **Spatial Localization** (**s**) estimates physical coordinates $\mathbf{s}_i$ relative to the observer.

### 3.3 LAYER 2: RELATIONAL INTEGRATION ($\mathcal{L}_2$)

$\mathcal{L}_2$ constructs a structured scene graph $\mathcal{G} = (\mathcal{O}, \mathcal{E})$ by integrating relationships. Without high-level logic, this layer focuses on **Attribute Binding** and **Metric Measurement**. The output is a set of relational facts $\mathcal{F}_{rel} = \{r_{ij}, \forall i, j\}$, transforming the list $\mathcal{O}$ into a connected graph.

**Intra-Object Binding:** To solve the binding problem, we define a query function $Q(o_i)$ that verifies if distinct features belong to the same entity. Given observed attributes $\mathbf{a}$, it regresses missing $\bar{\mathbf{a}}$:

$$P(\bar{\mathbf{a}} \mid \mathbf{a}, \mathbf{x}) \rightarrow \hat{\mathbf{a}}$$

This enables cross-modal inference, such as $c \rightarrow \tau, \mathbf{s}$ or $\mathbf{s}, \tau \rightarrow c$.

**Inter-Object Relations:** We define edges $\mathcal{E}$ as quantifiable interactions via operators $\psi_k$:

$$r_{ij} = \psi_k(o_i, o_j) \in \mathbb{R}^d$$

Relations capture physical and structural interactions: For **Environment-Event** interactions, physical modifications (e.g., reverb, occlusion) and perceptual interactions (e.g., masking) are widely observed; while for **Event-Event** interactions, spatiotemporal structure (e.g., overlap, precedence) and spectral similarity (e.g., harmonic relations) should be detected and measured.

### 3.4 LAYER 3: COGNITIVE REASONING ($\mathcal{L}_3$)

The apex layer $\mathcal{L}_3$ integrates atomic facts ($\mathcal{L}_1$) and the relational graph ($\mathcal{L}_2$) with the knowledge base $\mathcal{K}$. It performs probabilistic inference to derive high-level judgments $\mathcal{J}$:

$$\mathcal{J}^* = \underset{\mathcal{J}}{\arg\max}\, P(\mathcal{J} \mid \mathcal{G}, \mathcal{K})$$

This layer supports four advanced reasoning modes:

- **Abductive & Causal Inference:** Distinguishes correlation from causality (e.g., sequential impacts) and infers global context from local evidence (e.g., vehicle sounds $\rightarrow$ "traffic").
- **Deductive & Commonsense Validation:** Validates perception using physical constraints (e.g., material properties from impact) and detects semantic anomalies via environmental priors.
- **Inductive & Analogical Generalization:** Generalizes recurring structures (e.g., alternating vocals $\rightarrow$ "dialogue") and characterizes unknown sources via physical analogy (e.g., Doppler $\rightarrow$ motion).
- **Multi-hop Traversal & Metacognition:** Executes graph traversal for complex queries (e.g., temporal search) and performs metacognitive tasks like counterfactual reasoning (e.g., no response $\rightarrow$ "empty").

**Note.** The ASA framework defines objects as disentangled tuples $(c, \tau, \mathbf{s})$. However, solving the spatial-semantic binding problem is a prerequisite for holistic perception. Therefore, this paper specifically focuses on the integration of semantic ($c$) and spatial ($\mathbf{s}$) dimensions. We assume the input audio represents a coherent scene or event window, treating $\tau$ as an implicit context rather than an explicit prediction target in this stage.

## 4 METHODOLOGY

### 4.1 PROBLEM FORMULATION AND DESIGN PRINCIPLES

We address the task of conditional text generation from binaural audio input. Given a two-channel audio waveform $x \in \mathbb{R}^{B \times 2 \times T}$ and a natural language prompt $p$, the model's objective is to generate

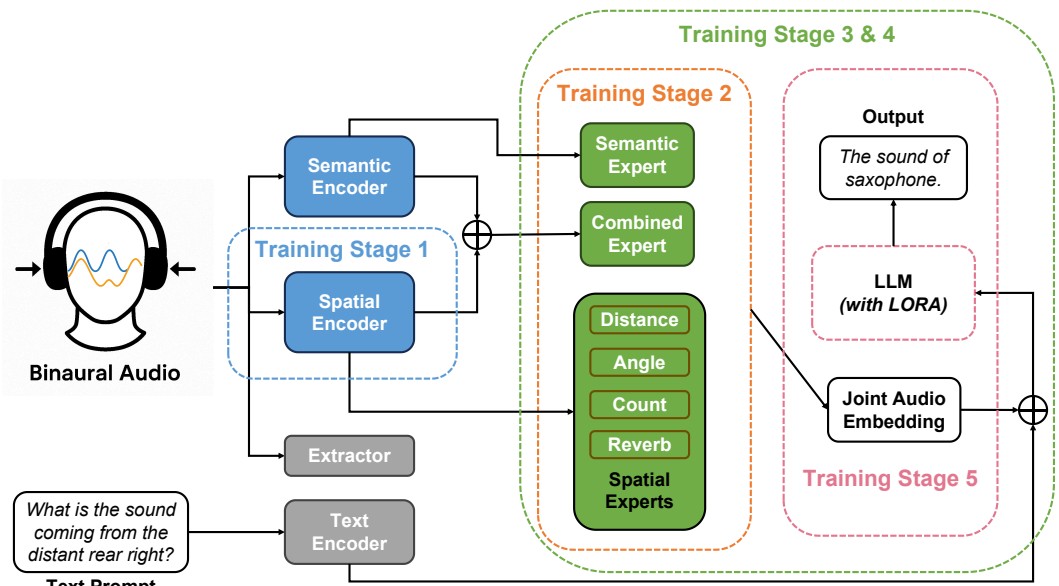

Figure 1: Model Architecture

a textual response $y$. To enable the model to comprehend not only the semantic content ("what") but also the spatial arrangement ("where") of the acoustic scene, we explicitly decompose the audio representation into two distinct components: a semantic embedding $z_{\text{sem}}(x)$ and a spatial embedding $z_{\text{sp}}(x)$. These representations are conditionally fused before being processed by the language decoder:

$$y \sim p_\theta\big( \cdot \, \big| \, \text{fuse}(z_{\text{sem}}(x),\, z_{\text{sp}}(x)),\, p\big).$$

Our framework is built upon three core design principles:

- **Decouple-and-Fuse.** Semantic and spatial attributes are modeled by independent, robust encoders. Their outputs are then integrated via a dense fusion mechanism. Unlike sparse routing approaches that may select partial information, our dense fusion ensures that the Language Model receives a complete, simultaneous view of both semantic identity and spatial location.

- **Minimally Invasive LLM Interfacing.** We avoid complex prompt templates and modifications to the LLM's native tokenization or positional encoding schemes. The final audio representation is simply appended to the prompt embeddings as a suffix, ensuring maximum compatibility with standard autoregressive frameworks.

- **Progressive Curricular Optimization.** We employ a multi-stage training curriculum that proceeds from representation learning to alignment and finally to policy optimization. This disentangles conflicting learning objectives and stabilizes the training of the composite model.

## 4.2 MODEL ARCHITECTURE

Our architecture, depicted in Figure 1, implements our "Decouple-and-Fuse" principle. The process begins as the input binaural audio is fed into two parallel backbones: a Semantic Encoder to extract a content representation ($W$) and a Spatial Encoder for locational cues ($S'$). These decoupled representations are then processed and integrated by our hybrid feature projector. This module employs a set of parallel expert pathways to project features into specialized physical subspaces, followed by a dense fusion layer that unifies them into a comprehensive audio embedding ($H_{\text{enc}}$). This embedding is appended to the prompt embeddings and passed to the LLM decoder for text generation.

**Semantic and Spatial Encoders.** For semantic feature extraction, we use a frozen Whisper encoder. To specifically isolate content, the binaural input is downmixed to mono by averaging the left and right channels. This process yields a semantic sequence $W \in \mathbb{R}^{B \times L \times 768}$.

For spatial features, we adapt the dedicated encoder from (Wu et al., 2025). It processes the complex STFT of the binaural signal to preserve phase information crucial for localization. The model alternates between frequency-band modeling and temporal modeling using self-attention. Critically, an attractor-based aggregation mechanism allows it to handle a variable number of sound sources. The final output is projected and resampled into a spatial feature sequence $S' \in \mathbb{R}^{B \times L \times 768}$, aligned with the semantic features.

**Hybrid Feature Projector.** The hybrid feature projector serves as the bridge between the decoupled representations and the decoder. It adopts a "split-transform-merge" strategy to maintain the integrity of specific acoustic attributes while providing a unified context to the LLM.

**Parallel Expert Pathways.** Instead of a generic linear projection, we employ a set of parallel feedforward networks, termed "Expert Pathways," to explicitly model distinct acoustic properties. The module comprises:

- **Semantic Pathway ($E_{\mathbf{sem}}$):** Processes the semantic representation $W$ to align content features with the LLM space.

- **Spatial Pathways ($\{E_{\mathbf{sp}}^{(k)}\}_{k=1}^4$):** Process the spatial representation $S'$. These pathways are specialized to extract distinct spatial attributes: direction, distance, room acoustics (reverberation), and source count. This explicit separation provides a strong inductive bias, ensuring that critical physical cues are preserved and highlighted before fusion.

**Dense Feature Fusion.** To avoid the information loss associated with sparse routing, we employ a dense fusion mechanism. The outputs of all expert pathways are concatenated to form a rich, multi-view representation, which is then projected by a fusion MLP into the final hidden state $H_{\mathrm{enc}}$. Formally, let $O_{\mathrm{sem}} = E_{\mathrm{sem}}(W)$ and $O_{\mathrm{sp}}^{(k)} = E_{\mathrm{sp}}^{(k)}(S')$. The fusion process is defined as:

$$H_{\mathrm{enc}} = \mathrm{MLP}_{\mathrm{fuse}} \left( \mathrm{Concat} \left[ O_{\mathrm{sem}} ; O_{\mathrm{sp}}^{(1)} ; O_{\mathrm{sp}}^{(2)} ; O_{\mathrm{sp}}^{(3)} ; O_{\mathrm{sp}}^{(4)} \right] \right) \in \mathbb{R}^{B \times L \times d_{\mathrm{dec}}}$$

This dense architecture ensures that for every audio token, the LLM has simultaneous access to the full spectrum of semantic and spatial information, facilitating the learning of complex cross-modal dependencies (binding).

**LLM Interface.** We employ the Qwen2-Audio 7B model as the language decoder. Its core parameters are kept frozen, and we use Low-Rank Adaptation (LoRA)(Hu et al., 2022) for parameter-efficient fine-tuning, applied only to the attention projection matrices (e.g., `q_proj`, `v_proj`). Following our principle of minimal intrusion, the audio hidden state $H_{\mathrm{enc}}$ is directly appended to the prompt embeddings $E(p)$, forming the input sequence $[\, E(p) \,;\, H_{\mathrm{enc}} \,] \in \mathbb{R}^{B \times (L_p + L) \times d_{\mathrm{dec}}}$. This simple and robust protocol eliminates the need for prompt engineering.

### 4.3 LEARNING: A PROGRESSIVE CURRICULUM FROM REPRESENTATION TO POLICY

To disentangle competing learning objectives, we adopt a five-stage curriculum. The input protocol and tokenization remain consistent across all stages to prevent training-inference mismatch.

**Stage 1: Encoder Pretraining.** In this preparatory stage, only the spatial encoder is trained. The objective is to learn a robust and general-purpose representation of spatial acoustics from a large-scale (potentially unlabeled) binaural audio dataset. This endows the encoder with a strong inductive bias for physical acoustic properties, decoupling this representation learning from the more complex cross-modal alignment task.

**Stage 2: Projector Alignment (Alignment I).** The audio encoders and the LLM are frozen, and only the hybrid feature projector (Expert Pathways and Fusion MLP) is trained. The optimization is driven by the standard cross-entropy loss $\mathcal{L}_{\mathrm{CE}}$ on the decoder output. The goal of this stage is to initialize the projection subspaces and align the fused audio representations with the pre-trained LLM's embedding space without altering the reasoning core.

**Stage 3: SFT 1.0 (Alignment II).** The encoders remain frozen, while the feature projector and the decoder's LoRA weights are trained jointly. The objective is to minimize $\mathcal{L}_{\text{CE}}$. This stage focuses on bridging the modality gap between the fused audio representations and the LLM's latent space, refining the cross-modal interface.

**Stage 4: SFT 2.0 (Formatting).** In this stage, the encoders remain frozen, while the projector and the decoder's LoRA weights continue to be trained jointly. The key change is the shift in data distribution towards chain-of-thought samples. The optimization is driven solely by the cross-entropy loss $\mathcal{L}_{\text{CE}}$. The purpose is to transition the model to end-to-end learning, fostering a co-evolution of the reasoning strategy and the desired output format (e.g., `<think>...</think><answer>...</answer>`).

**Stage 5: GRPO Preference Optimization.** Finally, only the LoRA weights are updated using GRPO. For each sample, we generate a group of $m$ candidate responses $\{y_i\}_{i=1}^m$ from the current policy $\pi_\theta$. Each response is scored by a reward function:

$$R(y) = R_{\text{fmt}}(y) + R_{\text{ans}}(y).$$

Here, $R_{\text{fmt}}$ assesses structural compliance (e.g., proper tag closure), while $R_{\text{ans}}$ measures correctness by parsing the content within the `<answer>` tag. GRPO uses the in-batch group mean as a baseline to compute the advantage:

$$A_i = \frac{R(y_i) - \frac{1}{m}\sum_{j=1}^m R(y_j)}{\text{std}_{j=1}^m (R(y_j))}$$

The policy is then updated by maximizing the following objective:

$$\mathcal{L}_{\text{GRPO}}(\theta) = \mathbb{E}_{(x,p),\{y_i\}\sim\pi_\theta}\left[\sum_{i=1}^m \log \pi_\theta(y_i|x,p) \cdot \text{stop\_gradient}(A_i)\right].$$

Keeping the encoders and projector frozen during this stage confines the optimization to the small set of LoRA parameters, ensuring stability and efficiency.

In summary, our methodology combines a decoupled representation with a progressive training curriculum to efficiently instill spatial reasoning into a pre-trained LALM.

## 5 EXPERIMENTS

This section details the experimental evaluation of our framework. We aim to validate the effectiveness of the Hybrid Feature Projector and the progressive curriculum in endowing LALMs with hierarchical spatial intelligence, moving from atomic perception ($\mathcal{L}_1$) to relational integration ($\mathcal{L}_2$) and complex reasoning ($\mathcal{L}_3$).

### 5.1 DATASETS AND SIMULATION PIPELINE

Lacking suitable public datasets for spatial audio reasoning, we developed a scalable simulation pipeline to generate data for our SFT and GRPO stages. The pipeline combines physically principled BRIR generation with dynamic scene synthesis. We construct diverse acoustic environments by varying room dimensions, wall absorption coefficients, and source positions. Detailed simulation parameters and geometric configurations are provided in Appendix A.

### 5.2 TRAINING SETUP

Our training regimen follows the progressive curriculum outlined in Section 4, beginning with Projector Alignment, followed by Supervised Fine-Tuning (SFT), and culminating in GRPO. We use LoRA for parameter-efficient fine-tuning and employ the AdamW optimizer. Complete hyperparameters, including learning rates, batch sizes, and hardware specifications, are detailed in Appendix B.

## 5.3 BENCHMARK CONSTRUCTION

To rigorously evaluate spatial reasoning, we constructed a 1,000-question multiple-choice benchmark using an LLM-powered pipeline. Crucially, the questions are designed to map directly to our proposed ASA framework:

- $\mathcal{L}_1$ **Perception (32.1%):** Tasks focusing on atomic identification, such as source counting, semantic recognition, and absolute localization.
- $\mathcal{L}_2$ **Integration (28.5%):** Tasks requiring attribute binding, such as associating a specific sound event with a specific location or linking environmental cues to source properties.
- $\mathcal{L}_3$ **Reasoning (39.4%):** Tasks involving high-level inference, such as counterfactual reasoning and summarizing the global scene context.

Detailed prompt templates for benchmark generation are provided in Appendix C.

## 5.4 MAIN RESULTS

To contextualize the performance of TWNM, we compare it against BAT (Zheng et al., 2025), a representative state-of-the-art baseline. Since BAT lacks the capability to generate structured chain-of-thought outputs, we adopted a hybrid evaluation protocol. For $\mathcal{L}_1$ Perception, we evaluated BAT using direct generation. For complex $\mathcal{L}_3$ Reasoning and $\mathcal{L}_2$ Integration tasks, we converted the multi-choice questions into boolean (True/False) discrimination tasks for each option and report the "Per-Option Accuracy". Note that for this binary metric, a random guess yields 50%, whereas standard MCQA accuracy for our model has a random baseline of 25%.

As shown in Table 1, the baseline BAT achieves only 36.40% on $\mathcal{L}_3$ tasks—well below the random chance threshold of 50%. This indicates that without explicitly decoupled spatial representations, generic LALMs suffer from systematic hallucinations. In contrast, our model, equipped with the Hybrid Feature Projector and Dense Fusion, achieves robust performance (79.60%) on the stricter MCQA metric, validating the necessity of our architecture for high-level reasoning.

Table 1: Comparative results and ablation study across ASA levels. Accuracy (%) is reported for the baseline (BAT) and each stage of our curriculum. BAT's performance on $\mathcal{L}_2$ and $\mathcal{L}_3$ is reported as Per-Option Belief Accuracy (random=50%), while ours is MCQA Accuracy (random=25%). The progressive curriculum yields significant gains, particularly in $\mathcal{L}_3$ Reasoning.

| Competency Area | BAT (Zheng et al., 2025) | SFT 1.0 | SFT 2.0 | GRPO |
|---|---|---|---|---|
| Perception ($\mathcal{L}_1$) | 24.57 | 40.72 | 62.36 | 61.05 |
| Integration ($\mathcal{L}_2$) | 35.52$^\dagger$ | **63.12** | 56.17 | 57.75 |
| Reasoning ($\mathcal{L}_3$) | 36.40$^\dagger$ | 43.64 | 66.89 | **79.60** |
| Overall Accuracy | 32.50 | 48.15 | 63.00 | **68.69** |

$\dagger$ Denotes Per-Option Accuracy (Binary T/F) where random baseline is 50%.

Our curriculum constitutes a staged ablation where each phase contributes a distinct capability. Moving from base alignment (SFT 1.0) to full instruction tuning (SFT 2.0) and finally to preference optimization (GRPO), overall accuracy rises from 48.15% to 68.69%.

**The Evolution of Reasoning ($\mathcal{L}_3$).** Preference optimization yields the largest gains on complex reasoning tasks, which jump from 43.64% (SFT 1.0) to 79.60% (GRPO). Specifically, the scene summarization task reaches 97.22% accuracy, and causal intent reasoning attains 92.00%. This confirms that while SFT 1.0 establishes the physical grounding, the combination of Chain-of-Thought (SFT 2.0) and GRPO is essential for unlocking high-level cognitive analysis.

**The Perception-Integration Trade-off.** We observe an interesting dynamic in $\mathcal{L}_2$ Integration tasks. SFT 1.0 achieves the highest performance (63.12%) in this category, outperforming subsequent stages. We interpret this as a trade-off: SFT 1.0 is optimized for direct, mechanical feature binding via the Dense Fusion mechanism. The transition to SFT 2.0 introduces complex language generation, which significantly boosts semantic perception ($\mathcal{L}_1$, 40.7% $\rightarrow$ 62.4%) and reasoning, but

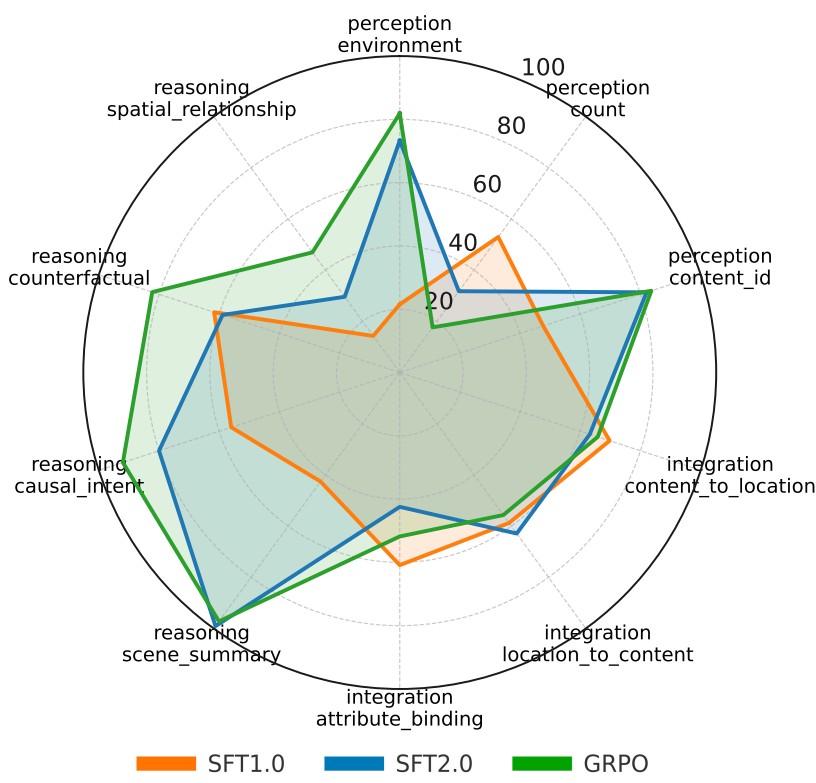

Figure 2: Task-wise accuracies on the ten benchmark categories for SFT1.0, SFT2.0, and GRPO.

incurs a temporary alignment tax on low-level binding precision ($63.1\% \rightarrow 56.2\%$). GRPO successfully recovers some of this capability ($56.2\% \rightarrow 57.8\%$) while maintaining peak reasoning performance.

**Remaining Challenges.** While the model excels at spatial semantics, the *Perception Count* task remains challenging, dropping to $17.65\%$ in the final stage. This suggests that while LALMs are excellent at qualitative spatial reasoning (e.g., "farther", "left of"), fine-grained numeracy remains an open challenge for end-to-end audio models.

## 6 CONCLUSION

In this work, we addressed the critical limitation of existing Large Audio-Language Models—their inability to perceive spatial audio—by introducing **The World is Not Mono (TWNM)**. Guided by our proposed hierarchical Auditory Scene Analysis (ASA) framework, this system bridges the gap between single acoustic tasks and cognitive reasoning. We demonstrated that a combination of large-scale synthetic binaural data, a hybrid feature projector utilizing dense fusion, and a progressive SFT-to-GRPO curriculum can successfully endow LALMs with robust spatial reasoning. Our experiments validated the superiority of the dense fusion mechanism over unified baselines, while preference optimization proved decisive for mastering complex reasoning. By moving beyond the mono paradigm, our work establishes a clear pathway for three-dimensional auditory intelligence. Future work will focus on enhancing the model's generalization to diverse in-the-wild recordings and extending our framework to support multi-channel formats like Ambisonics, further broadening its applications in embodied AI, augmented reality, and assistive technologies.

## 7 ETHICS STATEMENT

All authors of this paper have read and adhered to the ICLR Code of Ethics. This research does not involve human subjects, personally identifiable information, or sensitive data. The datasets utilized are publicly available and have been handled in accordance with their specified licenses. While we have strived to ensure the fairness and robustness of our methods, we acknowledge that potential biases may exist in the underlying data, which could be reflected in the model's performance. We believe the potential societal benefits of this work in advancing the understanding of audio scene analysis outweigh the foreseeable risks. We declare no conflicts of interest.

## 8 REPRODUCIBILITY STATEMENT

We are committed to ensuring the reproducibility of our research. All datasets used in our experiments are publicly available. For the review period, we include anonymized supplementary materials containing core implementation modules and configuration files to facilitate inspection of our method. A fully runnable codebase with scripts and instructions will be released upon acceptance.

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

APPENDIX

## A DATA SIMULATION DETAILS

To ensure the diversity and physical realism of our training data, we implemented a comprehensive simulation pipeline governed by the following parameters.

**Room Acoustics and Geometry.** We generate Binaural Room Impulse Responses (BRIRs) using the `pyroomacoustics` simulator (Scheibler et al., 2017). Rooms are modeled as shoebox shapes with dimensions sampled uniformly from three categories:

- **Small:** $x \in [3, 5]$ m, $y \in [4, 6]$ m, $z \in [2.5, 3.5]$ m

- **Medium:** $x \in [8, 12]$ m, $y \in [10, 15]$ m, $z \in [3, 5]$ m

- **Large:** $x \in [20, 30]$ m, $y \in [25, 35]$ m, $z \in [10, 15]$ m

Wall absorption coefficients $\alpha \in [0, 1]$ are sampled to simulate varying Reverberation Times (RT60):

- **High Reverberation (Reflective):** $\alpha \in [0.05, 0.25]$

- **Medium Reverberation:** $\alpha \in [0.25, 0.5]$

- **Low Reverberation (Absorptive):** $\alpha \in [0.5, 0.95]$

For each room, we randomly place one receiver and 30 candidate source positions. The receiver's Head-Related Transfer Functions (HRTFs) are sourced from the FABIAN dataset (Brinkmann et al., 2017).

**Source Material and Scene Synthesis.** Audio clips are sampled from FSD50K (Fonseca et al., 2022), Clotho v2 (Drossos et al., 2019), MusicCaps (Agostinelli et al., 2023a), and Emilia (He et al., 2024). These dry clips are convolved with the generated BRIRs, scaled by a random gain, and summed to create the final mixture. This pipeline ensures precise ground-truth labels for $\mathcal{L}_1$ (location, class) and $\mathcal{L}_2$ (spatial relations) tasks.

**Data Formats.** The training data evolves across stages:

- **SFT Data:** The initial stage employs open-ended QA pairs to align the Feature Projector with the LLM. The subsequent SFT 2.0 stage introduces Chain-of-Thought (CoT) formatting (using `<think>` and `<answer>` tags) to prime the model for reasoning.

- **GRPO Benchmark:** A held-out set of multiple-choice questions designed to probe specific spatial competencies, serving as the environment for reinforcement learning.

## B TRAINING IMPLEMENTATION DETAILS

**Hyperparameters.** We utilize the AdamW optimizer across all training stages. The detailed hyperparameter configuration for each stage of our progressive curriculum is provided in Table 2.

**System Configuration.** All experiments were conducted on a cluster of 8 NVIDIA RTX 5090 GPUs (32 GB VRAM each). We leverage `torchrun` for distributed training and utilize the Deep-Speed ZeRO Stage 2 strategy (Rajbhandari et al., 2020) with CPU offloading to maximize memory efficiency. The framework is implemented using PyTorch and the Hugging Face Transformers library.

Table 2: Hyperparameter settings for each training stage.

| Hyperparameter | Stage 2 (Alignment I) | Stage 3/4 (SFT) | Stage 5 (GRPO) |
|---|---|---|---|
| Learning Rate | $1 \times 10^{-4}$ | $5 \times 10^{-5}$ | $1 \times 10^{-6}$ |
| Batch Size (Global) | 128 | 128 | 64 |
| Epochs | 2 | 3 | 1 |
| Warmup Ratio | 0.03 | 0.03 | 0.1 |
| Gradient Clipping | 1.0 | 1.0 | 1.0 |
| LoRA Rank ($r$) | N/A | 8 | 8 |
| LoRA Alpha ($\alpha$) | N/A | 32 | 32 |
| Optimizer | AdamW | AdamW | AdamW |

## C    PROMPT FOR BENCHMARK GENERATION

---

**Prompt (for generating open-ended spatial QA)**

You are a top-tier AI course designer preparing graduation exam questions for an advanced spatial audio model. This model has already mastered the basics of recognizing sound content (semantic), localizing sources (localization), perceiving the environment (acoustics), and counting (count).

Your task: given a `<scene_description>`, design 1–2 **open-ended**, complex questions that require integrating multiple abilities, and provide detailed, fluent, high-quality answers.

**[Design Principles]**

- **Reject simplicity**: do not ask "Where is the dog?"; instead ask "What is the main activity in the scene, and where does it occur relative to me in space?"
- **Encourage reasoning**: propose questions that require inference from multiple cues. For example: "Based on the room's reverberation and the sounds inside, what is the most plausible type of place?"
- **Simulate dialogue**: both questions and answers should read like a natural conversation between humans.

**[Example]** `<scene_description>`

Indoors with slight reverberation, two sounds are audible. One is keyboard typing from straight ahead at a close distance. The other is birdsong coming from outside the window, located up-right and farther away.

`</scene_description>`

`<generated_data>`

```
[
  {
    "instruction": "Based on what you hear, can you infer where I might be and what I'm doing?",
    "answer": "It sounds like you're likely in a room—perhaps an office or at home. The clear, nearby
keyboard typing suggests you're working or studying on a computer. Meanwhile, the birdsong from
outside implies it's likely daytime and that you're close to a window."
  }
]
```

`</generated_data>`

**[Formal Task]** `<scene_description>`

`{{ insert RTSD here }}`

`</scene_description>`

`<generated_data>`

`</generated_data>`

---

## D    CASE STUDY

To further illustrate the reasoning ability of our model, we present a representative case study drawn from the evaluation set. The task requires the model to infer the most plausible explanation of an acoustic scene based on the perceived spatialized audio.

**Ground Truth (Reference)**

**Scene Description:**
– **Scene type:** Mixed, single sound source
– **Sound event:** Respiratory sound (breathing)
– **Azimuth:** 317°, **Elevation:** -13°, **Distance:** 17.4 m
– **Room acoustics:** Anechoic, absorption coefficient ≈ 0.99, RT60 = 0.0 s

**Correct Answer:**
A. The listener may be using headphones to hear a specially recorded or processed audio, in which a breathing sound is precisely placed at a specific position in the sound field.

**Question**

**Task:** Based on the audio you hear, which of the following explanations is the most reasonable for this scene?

- A. The listener may be using headphones to hear a specially recorded or processed audio, in which a breathing sound is precisely placed at a specific position in the sound field.

- B. In a wide-open space, someone is breathing about 17 meters away from the listener, and the openness of the space makes the sound unusually clear.

- C. The listener is in a completely silent room, and due to over-concentration, hallucinates the distant sound of breathing.

- D. This is a live recording in a large concert hall, where a breathing sound from an audience member was mistakenly amplified during post-production.

**Model Output (verbatim)**

`|<think>|` The core of the question is to infer the most reasonable scene based on the heard sound.
First, I analyze the key sound elements in the audio. I heard a clear breathing sound, coming from my front-left, about 17 meters away. The sound is very clear, almost without echo, which indicates I am in an open environment, such as a large space. The sound position is very precise, at azimuth 313 degrees and elevation 1 degree, which shows that the sound source is fixed rather than moving randomly.
Based on these observations, I evaluate each option:
– Option A states that the breathing sound was placed at a specific position. This matches exactly with the sound location I heard (azimuth 313 degrees). A processed audio, where a sound is placed at a specific position in the recording, is entirely possible.
– Option B describes a wide-open space with someone breathing. This contradicts the clear, echo-free sound characteristics. A wide-open space would produce noticeable reverberation.
– Option C suggests that the sound is a hallucination from silence. This is clearly wrong because I distinctly heard a breathing sound.
– Option D suggests a concert hall live recording where the breathing was over-amplified. However, a concert hall should have noticeable reverberation, which is absent here, so D is not reasonable.
In summary, Option A best fits the sound characteristics: a clear, precisely located breathing sound placed in a large space.`|</think>|`
According to the audio analysis, the sound comes from a precise location in a nearly echo-free large space, indicating that it is likely a processed audio rather than a real-world event.
`|<answer>| A |</answer>|`

# E    STATEMENT ON THE USE OF LARGE LANGUAGE MODELS (LLMS)

In the preparation of this manuscript, LLMs were utilized for two distinct purposes. First, as a general-purpose tool for tasks such as grammar correction and language polishing of the manuscript text. Second, as a core component of our methodology, LLMs were employed in a semi-automated

pipeline under human supervision to generate the questions and chain-of-thought rationales for our benchmark dataset, as detailed in Section 5.3. The core research ideation, experimental design, and the final analysis were conceived and executed by the authors. We confirm that the LLMs' role, while integral to the methodology, was that of a tool to implement our research design. The authors take full responsibility for all content presented in this paper, including the validity of the generated benchmark and any potential inaccuracies.

