# OpenReview forum: "The World is Not Mono: Enabling Spatial Understanding in Large Audio-Language Models"
_ICLR.cc/2026/Conference — Submitted to ICLR 2026_

### Official Review · Reviewer_fUwJ · 2025-10-27

**Soundness:** 2
**Presentation:** 2
**Contribution:** 2
**Rating:** 4
**Confidence:** 5

**Summary:**

This paper introduces "The World is Not Mono" (TWNM), a comprehensive framework for instilling spatial audio understanding in large audio-language models (LALMs). TWNM combines a synthetic large-scale binaural audio dataset, a Mixture-of-Experts (MoE) architecture that decouples semantic and spatial processing, and a multi-stage training curriculum culminating in reinforcement learning with Group Relative Policy Optimization (GRPO). The authors benchmark their system on an auto-generated, multi-task evaluation suite that probes perception, integration, and reasoning skills.

**Strengths:**

1. The paper tackles an often-overlooked limitation in LALMs—spatially aware auditory reasoning. The framing underscores the need for models capable of holistic scene analysis rather than mono-dimensional semantic understanding.
2. The use of decoupled semantic and spatial encoders combined via a conditional MoE is interesting.
3. Paper is well written and easy to follow.

**Weaknesses:**

1. This work lacks a proper baseline and only compares results from different training stages of its own model.

2. It would be better if the analysis also included sim-to-real performance.

3. The use of multiple encoders cannot guarantee complete disentanglement of audio information, and to some extent, it may even lead to desynchronization between different information streams. As discussed in the paper, the system may correctly identify sound sources A and B but confuse their directions or distances or others. The performance of spatial-relationship reasoning at 34.02% and attribute binding integration at 37.07% further prove this. Therefore, is such a disentangled multi-encoder approach combined with MoE truly an appropriate method?

**Questions:**

N/A

---

> ### Author Response · Authors · 2025-11-21
> **Response to Reviewer fUwJ: Baseline Comparisons, Sim-to-Real Constraints, and the Practicality of Decoupled Architectures**
>
> We sincerely thank the reviewer for their positive assessment of the paper's writing quality and for recognizing the significance of tackling holistic auditory scene analysis. We were particularly impressed by your insightful critique regarding the **"desynchronization"** risks in decoupled architectures (Weakness 3).
>
> We have addressed your three main concerns below:
>
> **1. Lack of Proper Baseline (W1)**
> * **Response:** We agree that a self-comparison is insufficient. In the revised manuscript (Section 6.2), we have added a comprehensive comparison with **BAT**, a representative spatial audio model.
> * **Comparison Results:**
>     * **Perception ($\mathcal{L}_1$):** BAT achieves only **$24.57\%$**, which is **poor** compared to our model ($61.05\%$). This demonstrates that generic spatial encoders struggle even with basic binaural attribute extraction in complex scenes.
>     * **Reasoning ($\mathcal{L}_3$):** BAT fails significantly on reasoning tasks, achieving only **$36.40\%$** (on a binary verification metric where random chance is $50\%$).
>     * **Ours:** Our model achieves **$79.60\%$** on reasoning (on a stricter 4-choice metric). This empirically proves that existing models lack the cognitive alignment required for complex spatial tasks.
>
> **2. Sim-to-Real Performance (W2)**
> * **Response:** We appreciate the suggestion. As detailed in our **General Response**, conducting a direct Sim-to-Real evaluation on public datasets (e.g., STARSS23) is currently scientifically infeasible for high-fidelity binaural reasoning.
>     * **Format Mismatch:** Real-world datasets are primarily **First-Order Ambisonics (FOA)**.
>     * **Physical Limitation:** Converting FOA to binaural audio results in severe information loss (theoretical reconstruction requires order $N \approx 32$, while FOA is $N=1$).
>     * **Perceptual Consequence:** Our tests show that FOA-rendered binaural audio suffers from severe **"in-head localization"** and spatial blur. Evaluating on such degraded data would measure conversion artifacts rather than the model's reasoning capabilities.
>
> **3. Decoupling vs. Desynchronization (W3 - Critical Point)**
> * **Response:** This is an exceptionally insightful comment. You correctly noted that multiple encoders *could* lead to desynchronization. However, we argue that a "Unified Encoder" is practically unrealistic and theoretically suboptimal for the following reasons:
>
>     * **The "Invariance-Equivariance" Dilemma:** Semantic recognition requires **Invariance** (e.g., a word should be recognized as the same regardless of its location), whereas spatial localization requires **Equivariance** (sensitivity to transformations like phase shifts). Forcing a single encoder to satisfy these opposing inductive biases simultaneously often degrades performance on both fronts.
>     * **Optimization Conflicts (Negative Transfer):** In multi-task learning, tasks with conflicting gradients can lead to **negative transfer**, where the optimization of one objective (e.g., semantics) dominates or destabilizes the other (e.g., spatial physics). Decoupling prevents this interference.
>     * **Neuroscientific Plausibility:** Our decoupled design mirrors the biological **"Dual-Stream Hypothesis"** in the auditory cortex, where the "What" (ventral) and "Where" (dorsal) pathways are physically separated before high-level integration.
>     * **Mitigation:** Regarding your concern about synchronization, we found that employing a **dense connection strategy** (concatenating features) rather than sparse routing helps mitigate this risk by encouraging the model to learn the joint distribution of the features.

---

> ### Author Response · Authors · 2025-11-25
>
> Dear Reviewer fUwJ,
>
> Thank you again for your valuable time and review.
>
> As we reach the midpoint of the rebuttal period, we are gently following up. We wanted to see if you had any remaining concerns after reading our response. We are, of course, ready to provide any additional clarification you might need.

---

### Official Review · Reviewer_UYn1 · 2025-10-30

**Soundness:** 2
**Presentation:** 2
**Contribution:** 1
**Rating:** 2
**Confidence:** 4

**Summary:**

This paper investigates equipping Large Audio Language Models (LALM) with spatial perception besides semantic understanding. To accomplish this, an MoE framework with four-stage learning is trained sequentially: (1) Learn the stem spatial audio representation with a spatial encoder, decoupled from the semantic representation; (2) Learn the MoE experts and router to process the concatenated semantic and spatial embeddings; (3) (SFT1.0) Align the MoE weights with LLM (finetuned with LoRA), with additional router supervision; and (4) (SFT2.0) Remove router supervision to further align MoE with LLM in end-to-end training. Additionally, GRPO is employed to further optimize the LLM under these spatial tasks. Moreover, this work proposes a new dataset and benchmark for LALM’s spatial audio reasoning by synthesizing binaural audio from single-channel audio data and querying LLM for question-answer pairs.

**Strengths:**

-	Originality: This paper accurately identifies the research gap in LALM’s spatial understanding capabilities. This is a nontrivial problem as humans have strong spatial cognition in hearing.

-	Quality:  The engineering efforts of this work are solid and well-planned. The four-stage training strategy appears justified with ablations, especially by the contrast of performance between SFT1.0 and SFT2.0.

-	Clarity: The proposed pipeline is illustrated clearly in a sequential manner.

-	Significance: This work is among the first to address the spatial audio understanding problem for LALMs.

**Weaknesses:**

-	The presentation of problem formulation is general and unclear. The authors define spatial audio understanding by examples in introduction, but do not categorize this understanding from a broad concept into specific tasks and define each. One can only grasp the outline of these tasks until the experiments section where dataset is introduced. Paragraph 2 in introduction is especially confusing because of this over-abstraction. Please see question 1 below for a request of clarification.

-	While it’s understandable that the proposed problem is relatively novel and lacks baseline models, sufficient experiments are still needed to demonstrate the proposed solution’s legitimacy. Instead of the complex training involved, one could prompt a spatial audio model for spatial localization, and another LALM for semantic understanding of the scene. Combining these predictions and prompting them to another LLM would resolve the “spatially deaf” limitation of current LALMs. Further experiments need to be conducted on these alternative baselines for comparison, delineating the necessity of the proposed method.

-	It’s curious why the proposed training pipeline requires this much complexity. From an engineering perspective, employing MoE is sound for its ability to dedicate parameter groups to semi-explicit subtasks. However, theoretically how much more gain can be achieved with MoE than a unified encoder-LLM mapper is under-studied here. It’s hard to justify this sheer amount of engineering tricks without seeing the performance tradeoff. The originality and novelty of this work thereby are heavily affected by the lack of this ablation.

**Questions:**

-	What are the major tasks in spatial audio understanding? Why is each dependent on binaural cues instead of single-channel semantics?

-	How much performance gain can be attributed to the fusing module? The model could be separately picking up semantic cues from the binaural channels to accomplish certain tasks.

-	Why MoE is needed to address this task? Could a unified encoder-LLM mapper achieve similar/better performance?

---

> ### Author Response · Authors · 2025-11-21
> **Response to Reviewer UYn1: Formalizing ASA, Theoretical Limitations of Pipeline Methods, and Architecture Ablation**
>
> We sincerely thank the reviewer for recognizing the **originality** of our work, particularly for affirming the importance of addressing "LALM Spatial Deafness." We take your criticisms regarding the unclear problem definition and architectural complexity very seriously. Your feedback has prompted a substantial reconstruction of both the theoretical framework and the model architecture in our revised manuscript.
>
> Below is our detailed response to your concerns and specific questions:
>
> 1. Clarification of Problem Definition: The Auditory Scene Analysis (ASA) Framework (W1 & Q1)
>
> **Response:** We fully accept the criticism that our initial formulation was too general. In the revised manuscript (Section 3.1), we introduce a hierarchical formal framework for **Auditory Scene Analysis (ASA)**, inspired by Bregman’s cognitive theory. This framework decomposes vague "spatial understanding" into three strictly dependent cognitive levels:
>
> * **$\mathcal{L}_1$ Static Identification:** Responsible for atomic perception. It extracts discrete attributes: *What* (semantic category $c$, extracted by the semantic encoder) and *Where* (spatial coordinates $\mathbf{s}$, extracted by the spatial encoder).
> * **$\mathcal{L}_2$ Relational Integration:** Responsible for solving the "Binding Problem"—locking specific semantic attributes to specific spatial locations (e.g., binding a "dog bark" to the "left" and a "cat meow" to the "right"). Furthermore, it models **physical interactions**, including Environment-Event interactions (e.g., timber coloration caused by reverberation) and Event-Event interactions (e.g., masking effects between sources).
> * **$\mathcal{L}_3$ Cognitive Reasoning:** Performs high-level logical inference based on the scene graph constructed in $\mathcal{L}_2$ (e.g., inferring motion direction via the Doppler effect, or performing counterfactual reasoning).
>
> 2. Why is the "Pipeline" Method (Discrete Models + LLM) Insufficient? (W2)
>
> **Response:** The reviewer suggested that the problem could be solved by having one spatial model output locations and another output semantics, combined via an LLM. While intuitively feasible, this approach suffers from fundamental theoretical limitations when handling complex spatial audio tasks:
>
> * **Information Bottleneck:** The pipeline approach forces intermediate representations through a low-bandwidth "text" interface (e.g., outputting "Dog at 30°, Cat at -30°"). This discretization discards the nuanced continuous acoustic features (such as phase interference and subtle reverb decay) necessary for judging object interactions. Tasks like determining "which sound is further away" or "whether sound A occludes sound B" rely on high-dimensional acoustic features, not highly compressed text labels.
> * **Temporal Desynchronization:** Real acoustic scenes are dynamic. If two independent models are used, guaranteeing strict "frame-level" alignment on a continuous audio stream is difficult. For example, when sources overlap temporally, an independent semantic model might output "Dog and Cat," while an independent spatial model outputs "Left and Right." The LLM cannot definitively resolve whether it is "Left Dog, Right Cat" or "Left Cat, Right Dog." Only an end-to-end joint feature space preserves this millisecond-level spatiotemporal correspondence to solve the binding problem.
> * **Error Propagation:** In a pipeline, if the front-end spatial localization model deviates due to noise, the back-end LLM cannot backtrack to the raw audio to correct the error. In our end-to-end architecture, the LLM always "sees" the raw acoustic features and can use contextual clues (e.g., semantic characteristics of the source) to help calibrate ambiguous spatial signals.
> * **Loss of Physical Context:** Simple combinations of discrete outputs ("Dog", "Left", "Bark") lose the rich physical context of *how* sounds interact. Complex acoustic phenomena—such as **spatial masking** (sounds obscuring each other based on location) or **environmental coupling** (how room geometry alters timbre)—cannot be captured by summing independent text predictions. An end-to-end joint representation is required to model these non-linear interactions.

---

> ### Author Response · Authors · 2025-11-21
> **Response to Reviewer UYn1 (Continue)**
>
> 3. Architecture Ablation: From Sparse MoE to Dense Fusion (W3, Q3)
>
> **Response:** Your suggestion regarding a "Unified Encoder-LLM Mapper" was highly insightful.
> * **We adopted this insight:** Inspired by your feedback, we performed a significant architectural upgrade. We removed the original sparse Gated MoE and adopted a **Hybrid Feature Projector with Dense Fusion** strategy.
> * **Why not a simple Linear Mapper?** While we shifted to dense connections (conceptually closer to a unified mapper), we retained **parallel expert pathways**. This is because semantic features (position-insensitive) and spatial features (highly position-sensitive) are orthogonal or even interfering in feature space. A simple linear map struggles to preserve both. Our "Parallel Processing + Dense Fusion" design achieves feature decoupling while ensuring the LLM receives a complete panoramic view at every step.
> * **Performance Gains & Trade-offs:** Experiments show that **Dense Fusion** consistently outperforms sparse MoE on the critical $\mathcal{L}_2$ binding tasks (avoiding information loss caused by routing errors). Although this increases inference computation (as all pathways must be calculated), the cost is justified to achieve robust "Semantic-Spatial Binding." This also explains why the relative gain of GRPO is smaller in the final version—the base architecture (Dense Fusion) now provides a much more stable starting point than the MoE.
>
> 4. Responses to Specific Questions
>
> **Q1: Why is every task dependent on binaural cues rather than monaural semantics?**
> * **Limitations of Mono:** We acknowledge that for a very small subset of simple tasks (e.g., identifying "what is this sound" or "is speech present"), monaural semantics are indeed sufficient.
> * **Necessity of Binaural:** However, for the "Where" tasks defined in $\mathcal{L}_1$ and the "Binding" tasks in $\mathcal{L}_2$, monaural audio is physically invalid.
>     * Monaural audio loses all directional information (Azimuth & Elevation), making it impossible to distinguish "Left" from "Right."
>     * More importantly, human hearing relies on **Spatial Release from Masking** effects generated by Interaural Time Differences (ITD) and Interaural Level Differences (ILD) to separate targets from overlapping sources. If only monaural audio is used, the spectra of overlapping sources undergo aliasing/mixing, making it impossible for the model to distinguish and bind multiple objects. Thus, spatial understanding strictly depends on binaural cues.
>
> **Q2: How much performance gain does the fusion module bring? Could the model extract semantic cues from binaural channels separately?**
> * **Clarification on "Separate Extraction":** The reviewer may assume we could encode Left and Right channels separately (e.g., two independent Encoders) and then merge them. We must clarify this based on acoustic principles: **Spatial information does not exist in a single channel, but in the differences (ITD/ILD) between channels.**
>     * If we perform semantic encoding on Left/Right channels separately (e.g., using Whisper), because most current semantic encoders are based on Magnitude Spectrograms, phase information is discarded, causing the microsecond-level ITD to be completely lost.
>     * Therefore, we must input binaural signals simultaneously during the feature extraction stage to preserve inter-channel phase relationships. There is currently no mature monaural pre-trained Encoder capable of perfectly preserving phase information.
> * **Gain from Fusion:** The core function of the fusion module is to achieve **Cross-Modal Binding**. The model does not process cues separately; rather, it must use the fusion module to learn the joint distribution between "semantic features" and "spatial features," thereby answering the question of "Who is Where."

---

> ### Author Response · Authors · 2025-11-25
>
> Dear Reviewer UYn1,
>
> Thank you again for your valuable time and review.
>
> As we reach the midpoint of the rebuttal period, we are gently following up. We wanted to see if you had any remaining concerns after reading our response. We are, of course, ready to provide any additional clarification you might need.

---

### Official Review · Reviewer_cEND · 2025-11-02

**Soundness:** 2
**Presentation:** 3
**Contribution:** 3
**Rating:** 4
**Confidence:** 3

**Summary:**

This paper introduces a framework called “The World is Not Mono (TWNM)”, which aims to make Large Audio-Language Models (LALMs) spatially aware. Most existing models only process mono audio, this paper tackles that by giving models the ability to reason about spatial cues like direction, distance, and room acoustics. The authors build a synthetic binaural dataset using physically simulated environments with BRIR and HRTF filters to generate spatially accurate audio scenes. They further propose a task-aware Mixture-of-Experts (MoE) architecture that separates semantic and spatial processing, with specialized experts for handling aspects like direction, distance, reverberation, and source count. The training follows a progressive curriculum, beginning with supervised fine-tuning and culminating in reinforcement learning through GRPO to enhance spatial reasoning. The paper presents a new spatial reasoning benchmark comprising 1,000 multiple-choice questions designed to test perception, integration, and reasoning capabilities. Overall, the model demonstrates strong improvements across all task types, achieving an overall accuracy of 61%, with particularly notable gains in complex reasoning tasks after GRPO training.

**Strengths:**

- The paper addresses a fundamental gap in LALMs — spatial reasoning, which few have tackled.
- The staged curriculum (SFT → SFT 2.0 → GRPO) helps stabilize training.

**Weaknesses:**

- Only one LALM i.e. Qwen2-Audio is tested.
- Limited experimental comparison with other spatial LALMs.
- Missing human evaluation.

**Questions:**

- Can this be generalized to other models like SALMONN, AudioGPT, or Whisper-based LALMs?
- Can spatial LALMs like BAT, etc. be compared?
- Have you tested on real spatial datasets (e.g., STARSS23 or L3DAS23)? If not, how do you expect it to handle real-world acoustics that deviate from simulation?

---

> ### Author Response · Authors · 2025-11-21
> **Response to Reviewer cEND: Generalizability, Baselines, and Real-World Adaptation**
>
> We sincerely thank the reviewer for recognizing the **originality** of our work in addressing the spatial reasoning gap in LALMs and for highlighting the effectiveness of our **staged curriculum**. We have carefully addressed your concerns below.
>
> **1. Generalizability to Other LALMs (Q1)**
> * **Response:** Yes, our framework is designed to be highly modular.
>     * **Architectural Modularity:** The core component, the **Hybrid Feature Projector** (updated from MoE in the revision), serves as an adapter bridging audio representations and the LLM. This module is independent of the specific LLM backbone.
>     * **Compatibility:** Methods like SALMONN (Vicuna-based) or AudioGPT also utilize a "Projector-to-LLM" paradigm. Our framework effectively replaces the generic projector with our dual-stream (Semantic+Spatial) projector and dense fusion mechanism.
>     * **Conclusion:** While we focused on Qwen2-Audio, the proposed **$\mathcal{L}_1 \to \mathcal{L}_3$ ASA framework** and **Projector Alignment** strategy are universally applicable to other Whisper-based LALMs.
>
> **2. Comparison with Spatial LALMs like BAT (Q2)**
> * **Response:** We have added a comprehensive comparison with **BAT** in the revised manuscript (Section 6.2).
> * **Why it was initially excluded:** We initially hesitated to include BAT because of a significant **format and capability mismatch**. BAT is designed for short-form QA and lacks the architectural capacity for the structured Chain-of-Thought (CoT) reasoning required by our $\mathcal{L}_3$ tasks. Comparing them directly required adapting BAT to a "binary verification" task (checking each option independently) while our model performed standard 4-choice MCQA.
> * **Comparison Results:**
>     * **Perception ($\mathcal{L}_1$):** BAT performs unsatisfactory results ($24.57\%$) and is outperformed by our model ($61.05\%$).
>     * **Reasoning ($\mathcal{L}_3$):** Crucially, BAT fails on reasoning tasks, achieving only **$36.40\%$** (worse than the random guess baseline of $50\%$ for binary verification).
>     * **Ours:** In contrast, our model achieves **$79.60\%$**, validating that explicit spatial decoupling is necessary for high-level cognition.
>
> **3. Sim-to-Real and Real-World Datasets (Q3)**
> * **Response:** As detailed in our **General Response**, we respectfully clarify why a direct Sim-to-Real evaluation on datasets like STARSS23 is currently infeasible for high-fidelity binaural reasoning.
>     * **The Core Issue:** Real-world datasets are primarily **First-Order Ambisonics (FOA)**. Converting FOA to binaural audio results in severe information loss (theoretically requiring HOA order $N \approx 30+$ for perfect reconstruction).
>     * **Perceptual Consequence:** Our internal tests confirm that FOA-rendered binaural audio suffers from severe **"in-head localization"** and spatial blur. Evaluating on such data would measure the conversion artifacts rather than the model's reasoning.
>     * **Future Work:** We plan to adapt our Spatial Encoder to natively accept Ambisonics (FOA/HOA) signals in future work.
>
> **4. Human Evaluation (W3)**
> * **Response:** While we did not conduct a dynamic user study, we performed a rigorous **human audit** of the benchmark itself.
>     * **Quality Assurance:** We manually reviewed the generated audio, metadata, and question logic in the test set to ensure the "Ground Truth" is physically and semantically correct.
>     * **Objective Proxy:** Since the evaluation consists of objective 4-choice questions (MCQA) based on this human-verified ground truth, the accuracy metrics serve as a reliable proxy for human evaluation. Furthermore, the `<think>` tags allow for qualitative human inspection of the reasoning process, as demonstrated in the Case Study.

---

> ### Author Response · Authors · 2025-11-25
>
> Dear Reviewer cEND,
>
> Thank you again for your valuable time and review.
>
> As we reach the midpoint of the rebuttal period, we are gently following up. We wanted to see if you had any remaining concerns after reading our response. We are, of course, ready to provide any additional clarification you might need.

---

### Author Response · Authors · 2025-11-21
**General Response: Theoretical Framework, Architecture Updates, and Sim-to-Real Justification**

We sincerely thank the reviewers for their constructive feedback. We are encouraged that the reviewers recognize the **originality** of tackling spatial reasoning in LALMs (Reviewer cEND, UYn1) and the **significance** of bridging the "spatial deafness" gap (Reviewer cEND, fUwJ).

In response to the reviewers' common concerns regarding **problem formulation**, **architectural validity**, **baselines**, and **Sim-to-Real evaluation**, we have performed a substantial revision. The key updates are summarized below:

**1. Formalized Theoretical Framework: Auditory Scene Analysis (ASA)**
* **Response to Reviewer UYn1:** We acknowledge that the initial formulation was too general. In the revision (Section 3.1), we have introduced a formal framework inspired by **Auditory Scene Analysis (ASA)**. We explicitly define three cognitive layers:
    * **$\mathcal{L}_1$ Static Identification:** Atomic perception of "what" (semantic) and "where" (spatial).
    * **$\mathcal{L}_2$ Relational Integration:** Solving the "binding problem" to associate attributes (e.g., linking a sound to a location).
    * **$\mathcal{L}_3$ Cognitive Reasoning:** High-level inference (causality, counterfactuals) based on the scene graph.

**2. Architectural Refinement: From Sparse MoE to Hybrid Dense Fusion**
* **Response to Reviewer fUwJ & UYn1:** Reviewer fUwJ raised a valid concern that multi-encoder decoupling might lead to "desynchronization" or binding errors. To address this, we upgraded the architecture from a sparse-gating MoE to a **Hybrid Feature Projector with Dense Fusion** (Section 4.2).
* Instead of dynamically selecting experts (which risks information loss), our improved architecture processes semantic and spatial streams through specialized parallel pathways and then employs a **dense fusion mechanism**. This ensures that the LLM receives a simultaneous, holistic view of all attributes at every step, significantly improving the **Attribute Binding ($\mathcal{L}_2$)** performance compared to the sparse routing approach.

**3. New Baseline Comparison: TWNM vs. BAT**
* **Response to All Reviewers:** We have added a comprehensive comparison with **BAT**, a representative SOTA spatial audio model (Section 6.2).
* **Key Finding:** While BAT performs comparably on basic perception, it fails catastrophically on **$\mathcal{L}_3$ Reasoning** tasks, achieving only **36.40%** accuracy (worse than random chance on binary verification). In contrast, our proposed model achieves **79.60%**, proving that generic audio encoders are insufficient for spatial intelligence.

**4. Clarification on Sim-to-Real Evaluation**
* **Response to Reviewer cEND & fUwJ:** We acknowledge the suggestion to test on real-world datasets (e.g., STARSS23). However, we respectfully clarify why a direct Sim-to-Real evaluation is currently infeasible for high-fidelity binaural reasoning:
    * **Data Format Mismatch:** Most public real-world spatial datasets record audio in **First-Order Ambisonics (FOA)**.
    * **Theoretical Limitation:** Converting FOA to binaural audio results in severe information loss. Acoustically, to avoid spatial aliasing and perfectly reconstruct the sound field for a human head up to 20 kHz, a spherical harmonic order of $N \approx kr \approx 32$ is theoretically required.
    * **Perceptual Consequence:** Our internal tests show that FOA-to-Binaural rendering suffers from severe **"in-head localization"** and spatial blur, making it impossible to evaluate fine-grained directional reasoning.
    * **Future Work:** We view support for native Ambisonics (FOA/HOA) input as a promising future direction to bridge this gap, rather than evaluating on degraded binaural conversions.

Reproducibility: As stated in our paper, we have uploaded the **anonymized core code** (including model architecture definitions and training scripts) in the supplementary material to facilitate reproducibility checks.

---

### Meta-Review · Area_Chair_iSJC · 2026-01-01

**Summary:**

The paper is tasked with advancing the spatial reasoning capabilities of large audio-language models, and presents a framework for semantic and spatial understanding of binaural audio using a fine-tuned large language model (Qwen2-audio). The paper segregates acoustic reasoning into three hierarchical levels: i) audio class and spatial cues, ii) acoustic relations integration, and iii) higher-level reasoning, including commonsense, analogical reasoning, and metacognition. A new synthetic QA dataset, called The World Is Not Mono (TWNM), consisting of 1,000 problems, is introduced to study reasoning at each of these levels. A curriculum- and GRPO-based training is presented to train a mixture-of-experts type model. Experiments against prior works such as BAT on the proposed dataset show that the GRPO approach leads to improvements.

**AC Comments:**

While the reviewers for this paper acknowledged the importance of the problem addressed and the proposed approach, they also raised important concerns on multiple fronts, namely:

i) the lack of strong baselines validating the claimed limitations of modern LALMs in spatial audio reasoning;

ii) the absence of ablation studies demonstrating the necessity of the elaborate mixture-of-experts model and its training schedule;

iii) the lack of real-world and sim-to-real experiments;

iv) the absence of human evaluations on the presented dataset; and

v) insufficient benchmarking of the TWNM dataset on state-of-the-art LALMs across the three reasoning levels.

In addition, the paper was significantly rewritten during the rebuttal phase, with new sections, new results, and a revised methodology; these changes alone would warrant a new review. Even after such major revisions, the paper still appears to require substantial further changes and is not ready for acceptance.

**Reviewer Concerns:**

*Reviewer cEND* points out that the paper considers only a single LALM (Qwen2-Audio), does not evaluate on real-world data, and lacks human evaluation. The challenges the paper claims current LALMs face in spatial audio reasoning are also not substantiated on the proposed dataset using standard benchmarks such as AudioGPT, SALMONN, etc.

*Reviewer UYn1* raises three important concerns: i) lack of clarity in the reasoning hierarchy in the initial submission; ii) lack of baseline performance that justifies the need for the proposed training pipeline; and iii) lack of ablation studies on the proposed model architecture and training schedule.

*Reviewer fUwJ* criticizes the paper for the lack of suitable baselines, the absence of real-world and sim-to-real experiments, and the lack of justification for using multiple encoders in the mixture-of-experts framework, which may even lead to desynchronization, as suggested by some of the empirical findings.

**Reviewer Scores:**

**Reviewer cEND:** The authors clarify that the proposed method is applicable to other Whisper-based LALMs, and the paper is extended with a new comparison to BAT, showing that the proposed approach performs substantially better. The authors also state that evaluation on real-world datasets is challenging because such datasets use first-order ambisonics, whereas the proposed model uses binaural audio; converting between the two may introduce significant errors. Further, they argue that human evaluation is unnecessary since dataset generation already included human verification.

[*AC’s take on the response*]
The authors have substantially revised the original submission with additional details and some comparison to prior work such as BAT. However, the lack of additional results requested by the reviewer (especially comparisons to SALMONN and AudioGPT) on the new dataset, the absence of reported performance on real-world datasets (even acknowledging possible conversion errors), and the lack of explicit human evaluation to establish the ideal performance achievable on the proposed dataset suggest that the reviewer would not have been fully satisfied by the response and would likely have maintained the original score.

**Reviewer UYn1:** To address concerns regarding the lack of clarity in the semantic hierarchy of reasoning problems, the authors revised the paper by introducing a new Section 3. The authors also argue that alternative approaches suggested by the reviewer may introduce several technical challenges that could lead to lower performance; however, no new results are provided to substantiate these claims. In addition, the authors made a major architectural change by discarding the previous sparse gated MoE and adopting a hybrid feature projector. While they report that the proposed dense fusion method outperforms the sparse MoE, they argue that separately encoding each binaural channel is theoretically suboptimal due to potential loss of synchronization.

[*AC’s take on the response*]
The paper appears to have undergone substantial revisions, including discarding the sparse MoE approach in favor of a hybrid design. Such major changes would warrant a thorough re-review in their own right. Nevertheless, the authors’ responses do not fully address the reviewer’s concerns, particularly regarding alternative design choices, justification for the complex training schedule, and the absence of ablation studies. As such, the reviewer would likely have retained a “reject” recommendation.

**Reviewer fUwJ:** The authors provide new results including an additional baseline (BAT), showing improved performance. Regarding the lack of real-world and sim-to-real experiments, the authors argue that such evaluations are difficult due to the first-order ambisonics-to-binaural conversion issue. They also claim that dense fusion can mitigate the desynchronization issue raised by the reviewer, although no empirical evidence is provided to support this claim.

[*AC’s take on the response*] Given that the authors do not provide new evidence beyond comparisons to BAT, it is unlikely that these largely unsubstantiated responses would have resulted in an increased reviewer score.

---

### Decision · Program_Chairs · 2026-01-26

Reject